# Underground Garage Patrol Based on Road Marking Recognition by Keras and Tensorflow

Jianwen Gan [1], Longqing Zhang [2] , Hongming Chen [3] , Liping Bai [1,*], Xinwei Zhang [2], Lei Yang [2] and Yanghong Zhang [2]

1 Department of Computer Science, Macau University of Science and Technology, Taipa, Macau, China
2 Department of Computer Science, Guangdong University of Science and Technology, Dongguan 523070, China
3 Key Laboratory of Oceanographic Big Data Mining & Application of Zhejiang Province, Zhejiang Ocean University, Zhoushan 316022, China
* Correspondence: lipbai@must.edu.mo

**Abstract:** The purpose of this study was to design an unmanned patrol service in combination with artificial intelligence technology to solve the problem of underground vehicle patrol. This design used the Raspberry Pi development board, L298N driver chip, Raspberry Pi camera, and other major hardware equipment to transform the remote control car. This design used Python as the programming language. By writing Python code, the car could be driven under the control of the computer keyboard and the camera was turned on for data collection. The Keras neural network library was used to quickly build a neural network model, the collected data was used to train the model, and the model was finally generated. The model was placed in the TensorFlow system for processing, and the car could travel in a preset track for unmanned driving.

**Keywords:** Keras neural network; TensorFlow; autopilot

## 1. Introduction

The overall plot ratio of urban residential quarters and actual population density is large. Developing ground parking in the city or building multistorey parking garages on the ground are not the best solutions for urban construction. Therefore, the construction of underground garages has become a trend for future development.

The safety of underground garages has not been paid enough attention, resulting in a high incidence of cases that occur in underground garages. The reasons for this include insufficient security equipment and factors such as inadequate management [1].

The topic of this study was mainly the design of an automatic driving patrol for underground garages, using the deep learning network to detect and identify preset roads, build a neural network model through the neural network library, and use the collected data to train the model. Finally, the model was generated and then processed using TensorFlow. The car could travel in a preset orbit to achieve unmanned garage patrol. At the same time, driving around the world is also a trend. The world's major technology companies are using artificial intelligence and driverless technology to develop unmanned vehicles with various functions, which not only saves considerable labor costs but also make people's lives more convenient and comfortable [2]. Unmanned driving mainly senses and recognizes the environment around the vehicle through on-board sensors and controls driving through the on-board computer.

## 2. Related Work

There are numerous road recognition research methods available today, the most common of which are in-vehicle vision systems and LIDAR (laser radar). LIDAR is not the preferred sensor for this task due to its high manufacturing costs and complicated

usage process. The focus of this paper was on image-based lane line and road marking detection algorithms. The edges, geometry, and texture of road markings are the most visible features in road marking detection, and they serve as the foundation for vehicle localization and navigation.

Traditional road marker detection methods are limited to a few scenarios. To detect lane lines, Mammeri et al. used MSER (maximum stable external region) and the progressive probabilistic Hough transform [3]. This method, however, is prone to being obstructed by obstacles and vehicles. To detect and fit straight lines, Huang et al. used an inverse perspective transform and feature voting mechanism [4]. The Kalman filter is one of the algorithms that is used to optimize and track lane line positions. However, the proposed method employs straight-line fitting for all lane lines, resulting in significant localization errors [5]. Niu et al. proposed a two-stage lane line detection method that extracted small line segments using a modified Hough transform (HT) and modeled the lanes using curve fitting to improve robustness [6]. It was difficult to detect and fit complex lanes at road intersections and ramps using this method.

Convolutional neural networks have demonstrated remarkable performance in picture classification and lane recognition as artificial intelligence has advanced. An end-to-end trainable network for lane line and road marker categorization in adverse weather situations was proposed by Lee et al. [7]. Neven et al. divided the segmentation task into two branches, lane detection and lane embedding, treating the lane detection problem as an instance segmentation problem [8]. Liang et al. suggested a CNN architecture with a novel prediction layer and scalable module called LineNet to produce high accuracy maps [9]. However, because their dataset based accuracy on GPS signals, the total inaccuracy was around 31.3 cm. Chen et al. proposed an alternative CNN mechanism for lane marker detection based on extracting lane marker features and used extended convolution to reduce the complexity of the algorithm, which primarily included semantic segmentation and post-processing [10]. All of the methods listed above provide good lane detection performance, but the majority of them are only intended for lane line detection and have high computational power requirements. The goal of this project was to propose an artificial intelligence-based unmanned patrol service that does not require a lot of arithmetic power [11]. Further, we combined the lightweight and convenient features of Raspberry Pi processors to create a cheap and convenient intelligent patrol solution for underground garages that can implement real-time road marking tasks on embedded systems and provide auxiliary information for more advanced autonomous driving.

The neural network in this design used Keras, an open-source neural network library written in Python. It can run on TensorFlow, Microsoft Cognitive Toolkit, Theano, or PlaidML. Users can build deep neural networks with Keras to quickly experiment with their own ideas and guesses [12]. At the same time, the modularity and scalability of Keras is easy for users to use. Keras contains many implementations of commonly used neural network building blocks, such as activation functions, optimizations, and a set of tools that make it easier to work with image and text data.

After the collected data were used to train the model, TensorFlow was used for processing. The predecessor of TensorFlow was DistBelief. DistBelief's function was to build a neural network-distributed learning and interaction system at various scales, also known as the "first-generation machine learning system." This project was developed and maintained by Google's artificial intelligence team, Google Brain. In November 2015, the development of the "Second Generation Machine Learning System," named TensorFlow and based on DistBelief, was completed and the code was open source [13]. TensorFlow has multiple projects, including TensorFlow Hub, TensorFlow Lite, and TensorFlow Research Cloud, and various application interfaces. TensorFlow is a symbolic math system based on data flow programming and an open-source software library with a multi-level structure that can be deployed on a variety of servers [14], PC terminals, and web pages and supports high-performance numerical computing, machine learning, and various perceptions and languages of understanding the task.

## 3. Hardware Construction

### 3.1. Raspberry Pi Camera Module

The camera is an important driving system for driverless driving. It is equivalent to the "eye" of a driverless car. It can acquire targets such as signal lights, road lines, pedestrians, etc., send the data to the visual processing part, and then use the processed results.

The Raspberry Pi camera module can be used to capture high-definition video as well as still photos. The camera consists of a small circuit board that is connected to the Raspberry Pi's Camera Serial Interface (CSI) bus connector via a flexible ribbon cable. The camera's image sensor has a native resolution of 5 megapixels and has a fixed focal length lens [15]. The camera software supports full-resolution still images with video resolutions of 1080p30 and 720p60. It can be used to take snapshots and as a home security camera.

The camera board is connected to the Raspberry Pi via a 15-pin cable. Only two connectors need to be connected, and the cable needs to be mounted to the camera board and the Raspberry Pi.

If not installed properly, the camera will not work. For the camera board, the blue mark at the end of the cable should be facing away from the board. In the Raspberry Pi section, the blue mark should be facing the direction of the network interface.

### 3.2. L298N Driver Chip

The L298N (Manufactured in Shenzhen, China by Jin Dapeng) is a driver IC that is an integrated monolithic circuit in a 15-pin Multiwatt. It is a bidirectional motor driver with a built-in L298 dual H-bridge motor driver that accepts standard TTL logic levels to drive inductive loads such as relays, solenoids, DC, and stepper motors [16]. Each of these H-bridges can provide 2 A of current, the power supply voltage range is 2.5–48 V, and the logic part is powered by 5 V, accepting 5 V TTL levels. Two enable inputs are provided to enable or disable the device independently of the input signal. The emitters of the lower transistors of each bridge are connected together and the corresponding external terminals can be used to connect the external sense resistors [17]. Additional power input is provided so that the logic operates at a lower voltage.

This design used the DuPont line to connect the front wheel motor of the car body to OUT3 and OUT4 of the L298N module, and the rear wheel motor of the car body was connected to OUT1 and OUT2 of the L298N module. Then, the IN control pin of the L298N module and the connection of the Raspberry Pi board were used [18].

The drive connection of the driver chip is shown in Table 1. When the signal of the enable terminal was 0, the motor was in a free stop state. When the enable signal was 1, if IN1 and IN2 were 00 or 11, the motor was in the braking state and the motor stopped rotating. If IN1 was 0 and IN2 was 1, motor A rotated clockwise; if IN1 was 1 and IN2 was 0, motor A rotated counterclockwise.

**Table 1.** Driver chip driver.

| ENA | IN1 | IN2 | The State of DC Motor A |
|:---:|:---:|:---:|:---:|
| 0 | X | X | Stop |
| 1 | 0 | 0 | Brake |
| 1 | 0 | 1 | Rotate Clockwise |
| 1 | 1 | 0 | Rotate Counterclockwise |
| 1 | 1 | 1 | Brake |

### 3.3. Raspberry Pi

Raspberry Pi is an ARM-based microcomputer motherboard with an SD/MicroSD card as the memory hard disk. There are 1/2/4 USB ports and a 10/100 Ethernet interface around the card motherboard, which can connect the keyboard, mouse, and network cable. At the same time, it has a TV output interface for video analog signals and an HDMI high-definition video output interface. All of the above components are integrated on a motherboard that is only slightly larger than a credit card. With all the basic functions of

a PC, only the TV and keyboard need to be turned on. It can perform functions such as spreadsheets, word processing, playing games, playing HD videos, and more.

### 3.4. Data Collection and Processing

The steering sensor generated a steering signal while the smart car was in motion, and the main control chip obtained that steering signal for output display. Instead of a single high level or low level trigger, the steering port outputted a period of around 2 s PWM waveform. We built a PWM detection algorithm to achieve steering signal acquisition because [19], at this time, we were unable to determine the level of the GPIO port to determine whether the automobile was turning. This detection algorithm is was named Algorithm 1.

---

**Algorithm 1:** Algorithm for turn signal detection method

---

**Input:** IO
**Output:** flag
  weight <- function (IO = 0;count < sizeof(IO);count++)
{
**Step1:** Count = 1
  Count = 1; Flag = 0
  SendMassage(flag)
**Setp2:** IO = 1
  Count = 0; flag = 1;
  First_flag = ON
  SendMassge(flag)
**End**

---

## 4. Software Design and Neural Network Training

### 4.1. Network Training

The neural network consists of individual neurons. In simple terms, each neuron performs some simple operations on the input to obtain the output [20]. Training a network adjusts the parameters in the network so that the output is close to the expected output of the learning sample.

A common neural network consists of an input layer, a hidden layer, and an output layer. Each layer consists of a number of neurons. The neurons of the input/output layer usually store only one value and do not perform operations themselves. The neurons of the hidden layer will operate on the input and pass the output to the next layer [21].

Figure 1 shows a diagram of the three-layer neural network's structure. The raw input information on the left is called the input layer, the neuron on the right is called the output layer (the output layer has only one neuron in the image), and the middle is called the hidden layer. The input layer receives input information, the hidden layer processes the input information, and the output layer outputs the processed information.

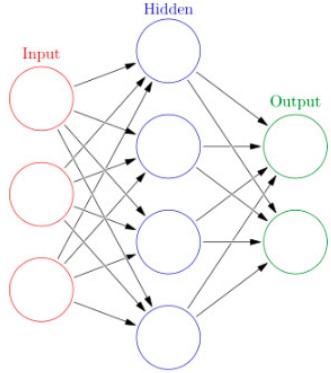

**Figure 1.** Three-layer neural network structure.

In this study, we created a neural network with three hidden layers containing ten neurons based on the 14 input attributes. The constructed neural network is shown in Figure 2.

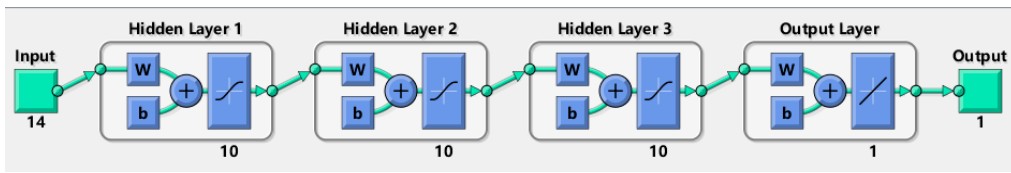

**Figure 2.** The neural network model constructed in this system.

The prediction model worked as follows.

Step 1. Input signals are $X_1, X_2, X_3, X_4, X_5, \ldots, X_{14}$

Step 2. The connection weights from the input layer to the hidden layer are $W_{kj}$, where $k$ denotes the number of neurons and $j$ denotes the number of hidden layer neurons. For example, $W_{14}$ represents the connection weight between the first neuron and the fourth node of the hidden layer.

*4.2. Neurons*

A neuron is a node of a neural network. Each neuron model has input, computational processing, and output capabilities. In the neuron network model shown in Figure 3, $x_1$ and $x_2$ are the input vectors. Weights are given when entering the input layer, and several input vectors are given several weights, such as $x_1w_1$ and $x_2w_2$. When the node z is reached, $x_1w_1$ and $x_2w_2$ are operated by the activation function g(z) [17], and finally the result a is output, and the operation of one neuron node is completed.

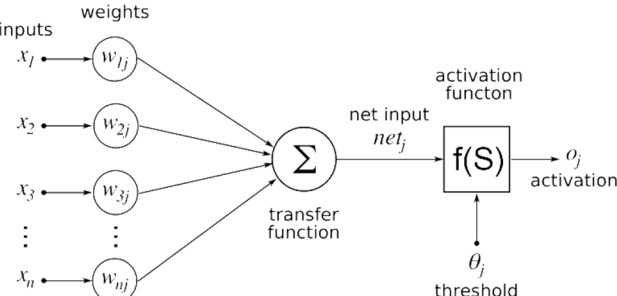

**Figure 3.** Neuron node model diagram.

*4.3. Convolutional Neural Network*

In deep learning, a convolutional neural network (CNN) is a type of deep neural network that is most commonly used to analyze visual images [22]. Convolutional networks are inspired by biological processes because the pattern of connections between neurons is similar to the organization of the animal's visual cortex [23]. Individual cortical neurons respond to stimuli only in restricted areas known as the receptive field. The receptive fields of different neurons partially overlap, enabling them to cover the entire field of view. Because artificial neural networks based on multi-layer supervised learning have good fault tolerance, adaptability, and weight sharing, they are widely used in image and video recognition, image classification, medical image analysis, and natural language processing [24].

In Figure 4 below, CONV is the convolution calculation layer, RELU is the excitation layer and an activation function, POOL is the pooling layer, and FC is the fully connected layer. Using the convolutional neural network model [25], the vehicle image was input into the neural network as a training set and the neural network parameters were corrected by multiple trainings [26]. Finally, a neural network model that could recognize the car picture was obtained.

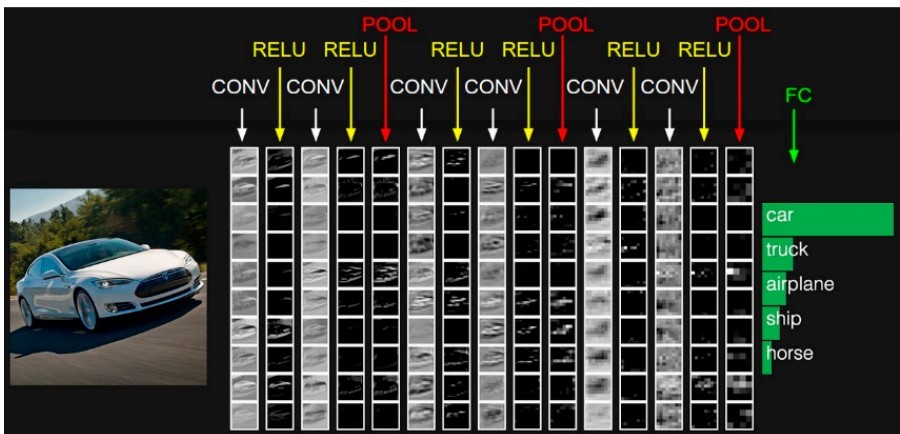

**Figure 4.** Convolutional neural network training process.

CONV: Extracts the local features of each image.
RELU: An activation function that converges quickly and requires a simple gradient.
POOL: Compresses the image and takes the average or maximum value of a region.
FC: Combines all calculation results to find the final answer.

### 4.4. Capture Images

This design established a connection with the Raspberry Pi through the computer, established a wireless link with the Raspberry Pi through the computer's VNC client, and then logged into the Raspberry Pi through the VNC client [27]. The data collection program (drive module and shooting module) was started in the Raspberry Pi. The keyboard was used to control the trolley on a predetermined track, and the predetermined track was repeatedly passed in a remote manner. Different commands were sent to the Raspberry Pi through the keyboard, and the Raspberry Pi output different levels of the control L298N motor to realize the travel of the unmanned smart car. Additionally, the camera was turned on during the journey to collect data captured during the trip. During this time, the thread was set, the direction key mark was recorded, and the mark was combined with the name of the picture to reset the name of the picture [28]. The pyGame module was used to detect the computer keyboard, control the movement of the car body through the computer keyboard, and set different keys for different control commands. The format method was used to name the image and place the key first in the name. Press a, turn left, mark 0; press d, turn right, mark 1; press w, forward to mark 2; press s, back, mark 3; press k, stop, marked as 4. The marked key was saved to the image and the corresponding direction was marked.

After collecting data many times, the data on the Raspberry Pi was transmitted to the computer because the Raspberry Pi has limited processing power, and the computing power of the computer was used to process the data. The data were run by preprocessing module program on the computer, the collected data were re-integrated, the name of the picture was first processed, the direction key mark of the picture was separated, the NumPy module was used to convert the data format from JPG to NPZ, and the direction markers of the image were stacked for subsequent data processing.

### 4.5. Data Processing

The data image was collected multiple times as a color image. For lane line detection, too much redundant or repeated information was included, and the image processing workload needed to be reduced using gradation processing [29]. Color is one of the factors that distinguish the lane. For color, there are different encoding methods, of which RGB, YUV, etc. are common. Since the color of the lane is different from the surrounding environment, we set thresholds for RGB and separated the useful information [29].

After the original image was processed by gradation, its gray dynamic range was small, and the gray level of the image could be improved by the transformation function. The transformation function is expressed as:

$$s = T(\mathbf{r}) = \int_0^r P_r(\omega)d\omega \tag{1}$$

where $T(r)$ represents the gray level and satisfies the condition $0 \leq T(r) \leq 1$, which is the probability density function of the continuous random variable.

The histogram equalization image enhancement principle adopted in this section can be expressed as the uniform distribution of the original image histogram cumulative distribution, which can increase the dynamic range of the image gray value and improve the image contrast, so that the specific information is highlighted. The histogram of grayscale reflects the probability of occurrence of each gray value of the image, so that the total number of gray levels of the image is L and the number of pixels whose gray value is $k$ is $n_k$. Then, the frequency is used instead of the probability value to be expressed as:

$$P_r(\mathbf{r}_k) = \frac{n_k}{n}(0 \leq \mathbf{r}_k \leq 1) \tag{2}$$

For the probability that the gray value is k and n is the sum of the number of image pixels, the discrete form of the transformation function of Equation (2) can be expressed as:

$$s_k = T(\mathbf{r}_k) = \sum_{j=0}^{k} \frac{n_j}{n} = \sum_{j=0}^{k} P_r(\mathbf{r}_j) \tag{3}$$

In order to enhance the image contrast, the image transformation function was used to perform grayscale mapping to improve the visual effect of the image. The processed image is shown in Figure 5.

$$g(x,y) = \begin{cases} \frac{f_1}{t_1}f(x,y) & 0 \leq f(x,y) \leq t_1 \\ \frac{f_2-f_1}{t_2-t_1}[f(x,y)-t_1]+f_1 & t_1 \leq f(x,y) \leq t_2 \\ \frac{(L-1)-f_2}{(L-1)-t_2}[f(x,y)-t_2]+f_2 & t_2 \leq f(x,y) \leq (L-1) \end{cases} \tag{4}$$

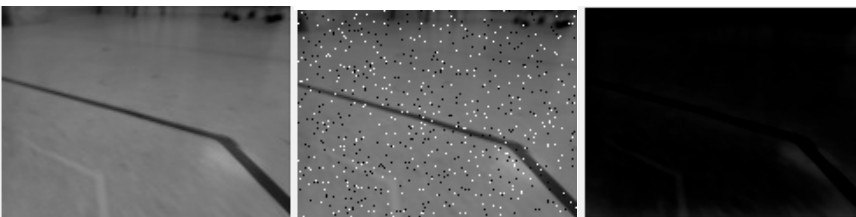

**Figure 5.** Processed image.

In Equation (4), $f(x, y)$ is the gray value of the original pixel, $g(x, y)$ is the function of the gray value of the pixel after stretching, and points $(t_1, f_1)$ and $(t_2, f_2)$ are the two inflection points of the pixel segmentation [28]. If different combinations of $t_1$, $t_2$, $t_3$, and $t_4$ are used, different stretching treatment effects will be obtained. The programmed image is shown in Figure 6.

### 4.6. Edge Detection

Edge detection is also a means of detecting lanes. Taking the grayscale image as an example, the grayscale value of each pixel is in the interval [0, 255], and the color of the lane is usually quite different from that of the road surface. We used the color change of the road surface to detect the lane [30].

The Canny edge detector is one of the edge detection methods. The general process of the Canny edge detector is as follows:

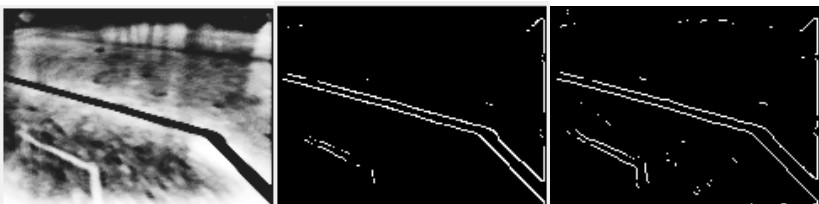

**Figure 6.** The programmed image.

Filter out any noise. The Gaussian filter is used for this purpose. An example of a Gaussian kernel of size = 5 that might be used is shown below:

$$K = \frac{1}{159} \begin{bmatrix} 2 & 4 & 5 & 4 & 2 \\ 4 & 9 & 12 & 9 & 4 \\ 5 & 12 & 15 & 12 & 5 \\ 4 & 9 & 12 & 9 & 4 \\ 2 & 4 & 5 & 4 & 2 \end{bmatrix} \tag{5}$$

Find the intensity gradient of the image. For this, we followed a procedure analogous to Sobel:

Apply a pair of convolution masks (in X and Y) directions:

$$G_x = \begin{bmatrix} -1 & 0 & +1 \\ -2 & 0 & +2 \\ -1 & 0 & +1 \end{bmatrix} G_y = \begin{bmatrix} -1 & -2 & -1 \\ 0 & 0 & 0 \\ +1 & +2 & +1 \end{bmatrix} \tag{6}$$

Find the gradient strength and direction with:

$$G = \sqrt{G_x^2 + G_y^2} \tag{7}$$

$$\theta = \arctan\left(\frac{G_y}{G_x}\right) \tag{8}$$

The direction is rounded to one of four possible angles (namely 0, 45, 90, or 135). Non-maximum suppression is applied. This removes pixels that are not considered to be part of an edge. Hence, only thin lines (candidate edges) will remain.

Hysteresis: The final step. Canny uses two thresholds (upper and lower):

If a pixel gradient is higher than the upper threshold, the pixel is accepted as an edge.

If a pixel gradient value is below the lower threshold, then it is rejected.

If the pixel gradient is between the two thresholds, then it will be accepted only if it is connected to a pixel that is above the upper threshold.

Canny recommended an upper: lower ratio between 2:1 and 3:1.

### 4.7. Convolutional Neural Network

A convolutional neural network is able to extract the characteristics of each picture, and when it has enough features, it can make relatively accurate judgments. The steps to construct a convolutional neural network are: construct a sequential neural network model, add a convolution kernel, add the Dropout() method to prevent overfitting, perform compression processing, return a one-dimensional array, and finally perform full join processing [31].

## 5. Training and Evaluation Models

Following sample processing of the data gathered by the smart cart, 70% of the photos produced were used as the training set and 30% were used as the test set. The neural network algorithm from Section 4 was then used for training prediction.

In this research, the learning rate of the model was increased and then decreased during the training process. In the first five epochs, the learning rate increased from 0 to 0.03 and then decreased at a rate of 0.0002/epoch. To optimize the gradient descent, the momentum optimization algorithm was used, with the momentum coefficient set to 0.8. During the patrol, the intelligent patrol car photographed the road markings. The convolutional neural network extracted features from each image, and a reasonably accurate judgment could be made once enough features were obtained. Constructing the convolutional neural network consisted of the following steps: building a sequential neural network model, adding a convolutional kernel, adding a Dropout method to randomly discard neural network units to prevent overfitting followed by a compression process to return a one-dimensional array, using the elu activation function in the experiment, and finally, a fully-connected process was obtained.

During the training of the neural network model, the parameters of the training model were first imported, including the preset model, learning rate, number of training, training set, and validation set data. The results of each training were kept using checkpoints, the training results were detected using early stop, the patience value was set to 4, the verbose value was set to 1, and the training was stopped when the model reached the optimal solution to prevent overfitting. Figure 7 illustrates the excellent accuracy of lane recognition in this experiment, which had an accuracy of 86.02%. To avoid overfitting, the training was stopped when the model reached the optimal solution. To avoid memory usage, the compiled neural network model was compiled using the fit generator block to read the data into memory. The VNC trained the neural network model program in the background and sent the generated model to the Raspberry Pi tensor flow system for processing after the operation. For unmanned patrol, the car could be driven on a pre-defined track.

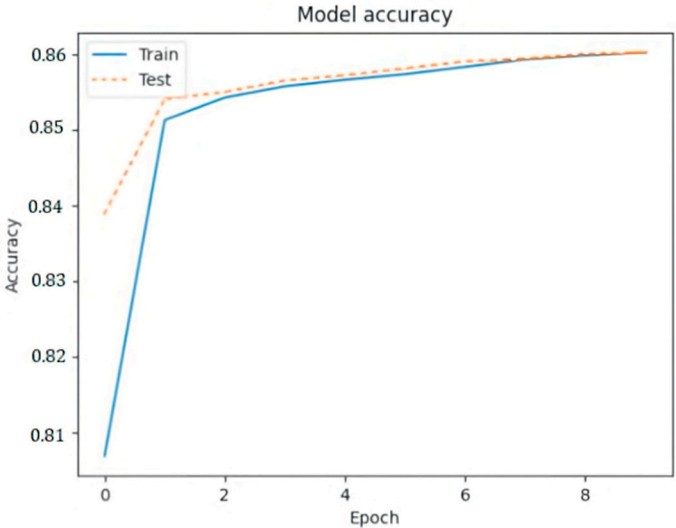

**Figure 7.** Accuracy Curve.

*Performance Metrics*

This work introduced the confusion matrix for the classification of curve detection in order to better explain the validity of the model chosen. The confusion matrix, sometimes referred to as the error matrix, is a matrix used to assess the classifier's classification accuracy. According to the intersection of the samples' true categories and classifier prediction categories, samples can be divided into four groups for classification: true positive (TP), false positive (FP), true negative (TN), and false-negative (FN). The relevant samples are

represented by TP, FP, TN, and FN. Classification models are typically assessed using characteristics such as accuracy, precision, recall, F1 value, and others. Table 2 displays the classification findings' confusion matrix.

**Table 2.** Confusion matrix of the classification.

| Actual | Predicted Positive | Predicted Negative |
|:---:|:---:|:---:|
| Positive | TP | FN |
| Negative | FP | TN |

Where TP is the proportion of samples with both positive actual and anticipated categories; FN represents the number of samples where the true category and forecasted category are both positive; FP is the proportion of samples where the anticipated category is positive but the actual category is negative; N = TP + FP + FN + TN is the total number of samples; and TN represents the number of samples where both the actual and projected categories are negative.

In this study, 920 images with positive and negative sample labels from the smart car's dataset were used to learn and validate lane recognition. According to the confusion matrix above, the accuracy, precision, and recall could be defined.

Accuracy is the correct proportion of all predictions and is defined as:

$$Accuracy = \frac{TP + TN}{TP + TN + FP + FN} \tag{9}$$

Precision is correctly predicted as the proportion of positive that is all positive and is defined as:

$$Precision = \frac{TP}{TP + FP} \tag{10}$$

Recall is correctly predicted as the proportion of positive that is all practically positive and is defined as:

$$Recall = \frac{TP}{TP + FN} \tag{11}$$

Another typical metric is the F-measure, which is the weighted average of precision and recall and is defined as:

$$\frac{1}{F_\beta} = \frac{1}{1 + \beta^2} \cdot \left( \frac{1}{P} + \frac{\beta^2}{R} \right) \tag{12}$$

The simplified formula is:

$$F_\beta = \frac{(1 + \beta^2) \times P \times R}{(\beta^2 \times P) + R} \tag{13}$$

In the formula, $\beta > 0$ indicates the relative importance of recall to precision. When $\beta = 1$, which is the standard F1 score, recall and precision are considered equally important. Furthermore, $\beta > 1$ indicates more emphasis on recall, whereas $\beta < 1$ indicates more emphasis on precision. In our research, the value for $\beta$ was 1. The *F*1 score combined the results of precision and recall. In this project, the average recall rate of lanes was 87.11% and the average precision was 89.12% when calculation of the aforementioned index formulas was also taken into account.

## 6. Result

The purpose of this design was to design an unmanned patrol service in combination with artificial intelligence technology to solve the problem of underground vehicle patrol. This design used the Raspberry Pi development board, L298N driver chip, Raspberry Pi camera, and other major hardware equipment to transform the remote control car. This

design used Python as the programming language. By writing Python code, the car could be driven under the control of the computer keyboard and the camera was turned on for data collection. The Keras neural network library was used to quickly build a neural network model, the collected data was used to train the model, and finally, the model was generated. The model was placed in the TensorFlow system for processing, and the car could travel in a preset track for unmanned driving.

**Author Contributions:** J.G., L.Z. and H.C. contributed equally to this work. Methodology, H.C.; Investigation, L.Y.; Resources, X.Z.; Data curation, J.G. and Y.Z.; Writing–original draft, L.Z.; Writing—review & editing, L.B. All authors have read and agreed to the published version of the manuscript.

**Funding:** This research was financially supported by Guangdong General University Young Innovative Talents (Project No. 2022KQNCX115), in part by Guangdong University of Science and Technology Quality Engineering (Project No. GKZLGC2022255), Special Projects in Key Areas for General Universities in Guangdong Province (No. 2021ZDZX1077), General Universities in Guangdong Province (No. 2021ZDZX1077), in part by National key research and development program of China, grant No. 2021YFF0700203, Natural Science Foundation of Guangdong Province of China (Grant No. 2020A1515010784), Guangdong General University Special Project in Key Areas (No. 2021ZDZX1077), Innovation and Improve School Project from Guangdong University of Science and Technology (No. GKY-2019CQYJ-3), and College Students Innovation Training Program held by Guangdong University of Science and Technology (No. 1711034, 1711080, and 1711088).

**Institutional Review Board Statement:** Not applicable.

**Informed Consent Statement:** Not applicable.

**Data Availability Statement:** Not applicable.

**Conflicts of Interest:** The authors declare that they have no conflicts of interest to report regarding the present study.

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
