# Peer review of "Underground Garage Patrol Based on Road Marking Recognition by Keras and Tensorflow"

_applsci, doi:10.3390/app13042385_

Round 1

Reviewer 1 Report

See attached file.

Author Response

Dear reviewer,

    According to the reviewer’s comments, we have revised the manuscript extensively. If there are any other modifications we could make, we would like very much to modify them and we really appreciate your help. We hope that our manuscript could be considered for publication in your journal. Thank you very much for your help.

Reviewer 2 Report

This manuscript aims to develop an unmanned patrol service through the rode line detection and the use of deep neural networks. The manuscript seems incomplete, and the novelty of this work is unclear. In addition, the following questions need further explanation.

1 The motivation and contribution of this work are unclear.

2 In the introduction part, please add more references to suppot your cliam, such as

(1) Bura, H., Lin, N., Kumar, N., Malekar, S., Nagaraj, S. and Liu, K., 2018, July. An edge based smart parking solution using camera networks and deep learning. In 2018 IEEE International Conference on Cognitive Computing (ICCC) (pp. 17-24). IEEE.

(2) Heimberger, M., Horgan, J., Hughes, C., McDonald, J. and Yogamani, S., 2017. Computer vision in automated parking systems: Design, implementation and challenges. Image and Vision Computing, 68, pp.88-101.

(3) Luo, C., Yu, L., Yan, J., Li, Z., Ren, P., Bai, X., Yang, E. and Liu, Y., 2021. Autonomous detection of damage to multiple steel surfaces from 360 panoramas using deep neural networks. Computer‐Aided Civil and Infrastructure Engineering, 36(12), pp.1585-1599.

3 Please do more literature reviews and summarize their limitations. 

4 The road line has been detected through edge detection. The purpose of using deep learning and the way to use deep learning are unclear.

5 The evaluation matrix for the proposed system is unclear

6. The details of the neural network, the parameters used to train the model, and the results need to be provided.

7 The conclusion is the same as the abstract. Please revise them

8 It seems that Figure 5 was not created by the authors. Please add reference for it

Author Response

Dear reviewer,

I hope that the changes I’ve made resolve all your concerns about the article. I’m more than happy to make any further changes that will improve the paper and/or facilitate successful publication.

Reviewer 3 Report

The authors aim to solve the problem of underground vehicle patrol by using AI approaches. This design uses the Raspberry Pi development board, L298N driver chip, Raspberry Pi camera and other major hardware equipment to transform the remote-control car. This design uses the Python language as the programming language. By writing Python code, the car can be driven under the control of the computer keyboard, and the camera is turned on for data collection. The Keras neural network library is used to quickly build a neural network model, and the collected data is used to train the model, and finally the model is generated. The model is placed in the TensorFlow system for processing, and the car can travel in a preset track for unmanned driving.

The paper is generally written. And some interesting results have been provided. However, this reviewer has the following comments:

The first three paragraphs in 1. Introduction should be combined where the research focus should be further enhanced. It is hard to see what do you want to express in the current context of the first three paragraphs. In general, each paragraph should has one theme.

Each abbreviation should be explained in its first usage.

The quality of Figure 2. Raspberry Pi and L298N driver chip connection and Figure 3. Three-layer neural network structure, should be improved.

The core techniques employed in the present should be clearly provided and analyzed. The authors just combined some AI techniques to solve the problem of underground vehicle patrol. Most of them are already existed in the literature. The main contributions of this work are quite vague. Furthermore, more details in the algorithm execution should be provided in the revision.

 Some more recent and related literature should be mentioned. And, please also check whether all the references are suitable. 

Author Response

Dear reviewer,

We sincerely thank the editor and all reviewers for their valuable feedback that we have used to improve the quality of our manuscript. The changes we made are in the word version of track change, and the word version has many modification marks.

Best regards,

Longqing zhang

Round 2

Reviewer 1 Report

See attached file.

Author Response

Dear reviewer,

We feel great thanks for your professional review work on our article. As you are concerned, there are several problems that need to be addressed. According to your nice suggestions, we have made extensive corrections to our previous draft, the detailed corrections are listed below.

Reviewer 1#:

  1. English. Please, carefully rewrite the paper so to be decently readable on its whole;
  2. Introduction.  It must be enlarged so to properly context the work, the problem it aims to solve, along with proposing a short hint of the solution;
  3. Related Works Section. A proper Related Works Section is missing, whose objective is to clearly show similarities and improvements with respect to the current state- of-the-art, and by clearly stating how this work advances it;
  4. The systems components are roughly described, but a proper description of the system, its working principle, and meaningful details are not present.  Please, bear in mind that, for instance, the fact the RPi has 40 pins is not so relevant from a scientific point of view. Unfortunately, the paper is full of such irrelevant comments;
  5. Neural Network.  While it is clear a CNN was adopted, no details on the architec- ture, the number of layers, the training procedure, the adopted dataset, etc... were provided. The paper is too vague under this perspective;
  6. Test.  Proper, methodological, and scientifically soundness are not included at all. How this system can be properly assessed?
  7. Conclusion. Similarly, since proper Tests are missing, a proper conclusion cannot be drawn (apart from the fact that a specific Session is missing).

Author's Answer

  1. we has undergone English language editing by MDPI. The text has been checked for correct use of grammar and common technical terms, and edited to a level suitable for reporting research in a scholarly journal.
  2. We have added content:

There are numerous road recognition research methods available today, the most common of which are in-vehicle vision systems and LIDAR (laser radar). LIDAR is not the preferred sensor for this task due to its high manufacturing costs and complicated usage process. The focus of this paper is on image-based lane line and road mark detection algorithms. The edges, geometry, and texture of road markings are the most visible features in road marking detection, and they serve as the foundation for vehicle localization and navigation.

Traditional road marker detection methods are limited to a few scenarios. To detect lane lines, Mammeri et al. used MSER (maximum stable external region) and the progressive probability hoff transform. This method, however, is prone to being obstructed by obstacles and vehicles. To detect and fit straight lines, Huang et al. used an inverse perspective transform and a feature voting mechanism. The Kalman filter is one of them, and it is used to optimize and track the lane line positions. However, their proposed method employs straight-line fitting for all lane lines, resulting in significant localization errors. J. Niu et al. proposed a two-stage lane line detection method that extracts small line segments using a modified Hough transform (HT) and models the lanes using curve fitting to improve robustness.The method is difficult to detect and fit complex lanes at road intersections and ramps.

Convolutional neural networks have demonstrated remarkable performance in picture classification and lane recognition as artificial intelligence has advanced. An end-to-end trainable network for lane line and road marker categorization in adverse weather situations was proposed by Lee et al. Neven et al. divided the segmentation task into two branches: the lane detection branch and the lane embedding branch, treating the lane detection problem as an instance segmentation problem. D. Liang et al. suggested a CNN architecture with a novel prediction layer and a scalable module called LineNet to produce high accuracy maps. However, because their dataset bases accuracy on GPS signals, the total inaccuracy is around 31.3 cm.P.R. Chen et al. proposed an alternative CNN mechanism for lane marker detection based on extracting lane marker features and used extended convolution to reduce the complexity of the algorithm, which primarily includes semantic segmentation and post-processing. All of the methods listed above provide good lane detection performance, but the majority of them are only intended for lane line detection and have high computational power requirements. The goal of this project is to propose an artificial intelligence-based unmanned patrol service that does not require a lot of arithmetic power, as well as to combine the lightweight and convenient features of Raspberry Pi processors to create a cheap and convenient intelligent patrol solution for underground garages that can implement real-time road marking tasks on embedded systems and provide auxiliary information for more advanced autonomous driving.

  1. In the section on related work we have made revisions
  2. In the fourth part we made modifications:

The neural network consists of individual neurons. In simple terms, each neuron performs some simple operations on the input to get the output[Islam, Chowdhury and Li et al (2018)]. Training a network adjusts the parameters in the network so that the output is close to the expected output of the learning sample.

Figure 3. Three-layer neural network structure.

A common neural network consists of an input layer, a hidden layer, and an output layer. Each layer consists of a number of neurons. The neurons of the Input/output layer usually store only one value and do not perform operations themselves. The neurons of the Hidden layer will operate on the input and pass the output to the next layer[Zhou, Li and Zhu et al. (2018)].

Fig.3 is a three-layer neural network structure diagram. The leftmost raw input information is called the input layer, the rightmost neuron is called the output layer (the output layer has only one neuron in the image above), and the middle is called the hidden layer. Input layer: receive input information; hidden layer: process input information; output layer: output processing information.

In this thesis, we create a neural network with three hidden layers containing ten neurons based on the 14 input attributes. The constructed neural network is shown in Figure 4.

Figure 4. The neural network model constructed in this system

The prediction model works as follows.

Step 1. Input signals are X1, X2, X3, X4, X5, …, X14

Step 2. The connection weights from the input layer to the hidden layer are Wkj, where k denotes the number of neurons and j denotes the number of hidden layer neurons. For example, W14 represents the connection weight between the first neuron and the fourth node of the hidden layer.

  1. Neural Network.  We have added content:

In this thesis, we create a neural network with three hidden layers containing ten neurons based on the 14 input attributes. The constructed neural network is shown in Figure 4.

Figure 4. The neural network model constructed in this system 

The prediction model works as follows.

Step 1. Input signals are X1, X2, X3, X4, X5, …, X14

Step 2. The connection weights from the input layer to the hidden layer are Wkj, where k denotes the number of neurons and j denotes the number of hidden layer neurons. For example, W14 represents the connection weight between the first neuron and the fourth node of the hidden layer.

  1. We have modified the contents:

In this research, the learning rate of the model is increased and then decreased during the training process. In the first five epochs, the learning rate increases from 0 to 0.03 and then decreases at a rate of 0.0002/epoch. To optimize the gradient descent, the momentum optimization algorithm is used, with the momentum coefficient set to 0.8. During the patrol, the intelligent patrol car photographs the road markings. The convolutional neural network extracts features from each image, and once enough features are obtained, a reasonably accurate judgment can be made.Constructing a convolutional neural network consists of the following steps: building a sequential neural network model, adding a convolutional kernel, and adding a Dropout method to randomly discard neural network units to prevent overfitting, followed by a compression process to return a one-dimensional array, using the elu activation function in the experiment, and finally a fully-connected process..

  1. We have modified the contents:

The purpose of this design is to design an unmanned patrol service in combination with artificial intelligence technology to solve the problem of underground vehicle patrol. This design uses the Raspberry Pi development board, L298N driver chip, Raspberry Pi camera and other major hardware equipment to transform the remote control car. This design uses the Python language as the programming language. By writing Python code, the car can be driven under the control of the computer keyboard, and the camera is turned on for data collection. The Keras neural network library is used to quickly build a neural network model, and the collected data is used to train the model, and finally the model is generated. The model is placed in the TensorFlow system for processing, and the car can travel in a preset track for unmanned driving.

Reviewer 2#:

No further comment. The revision can be accepted now.

Reviewer 3#:

No further comment. The revision can be accepted now.

Thank you very much for your attention and time.Look forward to hearing from you.

Yours sincerely,

Authors

Reviewer 2 Report

No further comment

Author Response

Dear reviewer,

We feel great thanks for your professional review work on our article. And we has undergone English language editing by MDPI. The text has been checked for correct use of grammar and common technical terms, and edited to a level suitable for reporting research in a scholarly journal.

Reviewer 2#:

No further comment. The revision can be accepted now.

Thank you very much for your attention and time.Look forward to hearing from you.

Yours sincerely,

Authors

Reviewer 3 Report

No further comment. The revision can be accepted now.

Author Response

Dear reviewer,

We feel great thanks for your professional review work on our article. And we has undergone English language editing by MDPI. The text has been checked for correct use of grammar and common technical terms, and edited to a level suitable for reporting research in a scholarly journal.

Reviewer 3#:

No further comment. The revision can be accepted now.

Thank you very much for your attention and time.Look forward to hearing from you.

Yours sincerely,

Authors

Round 3

Reviewer 1 Report

The paper slightly improved after its revision. However, it cannot be published in its present form since many major points must be fixed. Moreover, not all of my comments of the last previous round were properly taken into account. Therefore, I am proposing for major revision as mu judgment. The list of comment is below.

1. In the Related Works Section the references to the cited papers are missing.

2. Concerning comment #4 of the last review round, the Authors claim to have modified Section 4. However, the cited paragraph in the cover letter can be found both in the first and in the second versions of the paper. Asserting to have done a modification, and then not having actually done it is quite disrespectful. How can the Authors solve such comment? For the sake of convenience, I copy the comment below, and I strongly suggest the Authors to properly address it.

  • The systems components are roughly described, but a proper description of the system, its working principle, and meaningful details are not present. Please, bear in mind that, for instance, the fact the RPi has 40 pins is not so relevant from a scientific point of view. Unfortunately, the paper is full of such irrelevant comments;

3. Concerning comment #6, the added part is fine since it gives more details. However, such comment remains unsolved. I remind the Authors that without a proper and soundness test methodology the paper can never be published since the proposed solution cannot be rigorously assessed. Once again, for the sake of convenience, I copy the comment below, and I strongly suggest the Authors to properly address it.

  • Test. Proper, methodological, and scientifically soundness are not included at all. How this system can be properly assessed?

Author Response

Dear reviewer,

We appreciate all of your hard work reviewing our paper in a professional manner. There are a number of issues that need to be resolved, as you are aware. We have significantly revised our earlier draft in response to your thoughtful ideas; the full adjustments are shown below.

Reviewer 1#:

  1. In the Related Works Section the references to the cited papers are missing.

  1. Concerning comment #4 of the last review round, the Authors claim to have modified Section 4. However, the cited paragraph in the cover letter can be found both in the first and in the second versions of the paper. Asserting to have done a modification, and then not having actually done it is quite disrespectful. How can the Authors solve such comment? For the sake of convenience, I copy the comment below, and I strongly suggest the Authors to properly address it.

The systems components are roughly described, but a proper description of the system, its working principle, and meaningful details are not present. Please, bear in mind that, for instance, the fact the RPi has 40 pins is not so relevant from a scientific point of view. Unfortunately, the paper is full of such irrelevant comments;

  1. Concerning comment #6, the added part is fine since it gives more details. However, such comment remains unsolved. I remind the Authors that without a proper and soundness test methodology the paper can never be published since the proposed solution cannot be rigorously assessed. Once again, for the sake of convenience, I copy the comment below, and I strongly suggest the Authors to properly address it.

Test. Proper, methodological, and scientifically soundness are not included at all. How this system can be properly assessed?

Author's Answer

  1. we have modified.
  2. We removed this content:

The camera board is connected to the Raspberry Pi via a 15-pin cable. Only two connectors need to be connected, and the cable needs to be mounted to the camera board and the Raspberry Pi.

If it is not installed properly, the camera will not work. For the camera board, the blue mark at the end of the cable should be facing away from the board. In the Raspberry Pi section, the blue mark should be facing the direction of the network interface [Rajagopal, K. and Balakrishnan, S. (2016)].

3.2. L298N driver chip

The L298N is a driver IC that is an integrated monolithic circuit in a 15-pin Multiwatt and PowerSO20 package. It is a bidirectional motor driver with built-in L298 dual H-bridge motor driver that accepts standard TTL logic levels to drive inductive loads such as relays, solenoids, DC and stepper motors. Each of these H-bridges can provide 2A of current, the power supply voltage range is 2.5-48v, and the logic part is powered by 5v, accepting 5v TTL levels. Two enable inputs are provided to enable or disable the device independently of the input signal. Two enable inputs are provided to enable or disable the device independently of the input signal. The emitters of the lower transistors of each bridge are connected together and the corresponding external terminals can be used to connect external sense resistors[Shen, Hashimoto  and Matsuda et al (2019)]. Additional power input is provided so that the logic operates at a lower voltage.

This design uses the DuPont line to connect the front wheel motor of the car body to OUT3 and OUT4 of the L298N module, and the rear wheel motor of the car body is connected to OUT1 and OUT2 of the L298N module. Then use the IN control pin of the L298N module and the connection of the Raspberry Pi board[Wang, Chai and Nguyen (2019)].

The drive connection of the driver chip is shown in Tab. 1. When the signal of the enable terminal is 0, the motor is in a free stop state. When the enable signal is 1, if IN1 and IN2 are 00 or 11, the motor is in the braking state and the motor stops rotating. If IN1 is 0 and IN2 is 1, motor A rotates clockwise; if IN1 is 1 and IN2 is 0, motor A rotates counterclockwise.

Table 1. Driver chip driver.

ENA

IN1

IN2

The state of DC Motor A

0

X

X

Stop

1

0

0

Brake

1

0

1

Rotate Clockwise

1

1

0

Rotate Counterclockwise

1

1

1

Brake

3.3. Raspberry Pi

Raspberry Pi is an ARM-based microcomputer motherboard with SD/MicroSD card as the memory hard disk. Its physical picture is shown in Fig. 1. There are 1/2/4 USB ports and a 10/100 Ethernet interface around the card motherboard, which can connect keyboard, mouse and network cable. At the same time, it has a TV output interface for video analog signals and an HDMI high-definition video output interface. All of the above components are integrated on a motherboard that is only slightly larger than a credit card. With all the basic functions of a PC, you only need to turn on the TV and keyboard. Perform functions such as spreadsheets, word processing, playing games, playing HD videos, and more.

Figure 1. Raspberry Pi development board.

It has 40 GPIO ports, 4 USB 2.0 ports, power consumption of 4.0W, 1GB RAM and 64-bit 4-core 1.2GHz CPU. Compared with the 51 MCU and STM32 control processors, in addition to the same IO pin control, you can create a keyboard, mouse, monitor, power supply and micro SD card with Linux Distribution. A fully-fledged computer that runs applications from word processors and spreadsheets to games, enables more complex task management and scheduling and supports the development of higher-level applications[Sitawarin, Bhagoji and Mosenia et al (2019)].

3.4. Hardware Components Connection

There are two rows of interfaces, a total of 40. We define the pins of the Raspberry Pi as shown in Tab. 2.

Table 2. Raspberry Pi pin function table.

Function

Pin

Pin

Function

3.3V

1

2

5V

SDA.1

3

4

5V

SCL.1

5

6

GND

CPIO.7

7

8

TXD

GND

9

10

RXD

GPIO.0

11

12

GPIO.1

GPIO.2

13

14

GND

GPIO.3

15

16

GPIO.4

3.3V

17

18

GPIO.5

MOSI

19

20

GND

The upper row is even and the lower row is odd. This design selects the 7 GPIO ports of Raspberry Pi 7, 11, 13, 15 to connect with the logic input port of L298N driver chip, and ports 12 and 16 are connected with the enable port of L298N driver chip. Among them, ports 13 and 15 control the front wheel motor of the car, and port 16 serves as the enable end for the ports 13 and 15. Ports 7 and 11 controls the rear wheel motor of the car, and port 12 serves as the enable end for ports 7 and 11. Raspberry Pi connected L298N drive motor as shown in Fig. 2.

Figure 2. Raspberry Pi and L298N driver chip connection.

.

Our revised content is as follows:

3.3. Raspberry Pi

Raspberry Pi is an ARM-based microcomputer motherboard with SD/MicroSD card as the memory hard disk. There are 1/2/4 USB ports and a 10/100 Ethernet interface around the card motherboard, which can connect keyboard, mouse and network cable. At the same time, it has a TV output interface for video analog signals and an HDMI high-definition video output interface. All of the above components are integrated on a motherboard that is only slightly larger than a credit card. With all the basic functions of a PC, you only need to turn on the TV and keyboard. Perform functions such as spreadsheets, word processing, playing games, playing HD videos, and more.

3.4. Data collection and processing

The steering sensor will generate a steering signal while the smart car is in motion, and the main control chip will obtain that steering signal for output display. Instead of a single high level or low level trigger, the steering port will output a period of around 2 seconds PWM waveform. We have built a PWM detection algorithm to achieve the steering signal acquisition because at this time we are unable to determine the level of the GPIO port to determine whether the automobile is turning. This detection algorithm is known as Algorithm1.

Algorithm1:Algorithm for turn signal detection method

Input:IO

Output:flag

weight <- function (IO = 0;count<sizeof(IO);count++)

{

Step1:Count = 1

Count = 1;Flag = 0

SendMassage(flag)

Setp2:IO = 1

Count = 0;flag = 1;

First_flag = ON

SendMassge(flag)

End

  1. In the fivepart we made modifications:

Training and Evaluation Models

Following sample processing of the data gathered by the smart cart, 70% of the photos produced are used as the training set and 30% are used as the test set. The neural network algorithm from Chapter 4 is then used for training prediction.

In this research, the learning rate of the model is increased and then decreased during the training process. In the first five epochs, the learning rate increases from 0 to 0.03 and then decreases at a rate of 0.0002/epoch. To optimize the gradient descent, the momentum optimization algorithm is used, with the momentum coefficient set to 0.8. During the patrol, the intelligent patrol car photographs the road markings. The convolutional neural network extracts features from each image, and once enough features are obtained, a reasonably accurate judgment can be made.Constructing a convolutional neural network consists of the following steps: building a sequential neural network model, adding a convolutional kernel, and adding a Dropout method to randomly discard neural network units to prevent overfitting, followed by a compression process to return a one-dimensional array, using the elu activation function in the experiment, and finally a fully-connected process.

During the training of the neural network model, the parameters of the training model are first imported (preset model, learning rate, number of training, training set, and validation set data), the results of each training are kept using checkpoints, the training results are detected using early stop, the patience value is set to 4, the verbose value is set to 1, and the training is stopped when the model reaches the optimal solution to prevent overfitting. Figure 7 illustrates the excellent accuracy of lane recognition in this experiment, which has an accuracy of 86.02%. To avoid overfitting, the training is stopped when the model reaches the optimal solution. To avoid memory usage, compile the compiled neural network model using the fit generator block to read the data into memory. The VNC trains the neural network model program in the background and sends the generated model to the Raspberry Pi tensor flow system for processing after the operation. For unmanned patrol, the car can be driven on a pre-defined track.

Figure 7. Accuracy Curve

5.1 Performance metrics

This work introduces the confusion matrix for the classification research of curve detection in order to better explain the validity of the model chosen. The confusion matrix, sometimes referred to as the error matrix, is a matrix used to assess the classifier's classification accuracy. According to the intersection of the samples' true categories and classifier prediction categories, the samples can be divided into four groups for the classification: true positive (TP), false positive (FP), true negative (TN), and false-negative (FN). The relevant samples are represented by TP, FP, TN, and FN. Classification models are typically assessed using characteristics such as accuracy, precision, recall, F1 value, and others. Table 1 displays the classification findings' confusion matrix.

Table 1. Confusion matrix of the classification

Actual

Predicted Positive

Predicted Negative

Positive

TP

FN

Negative

FP

TN

Where TP is the proportion of samples with both positive actual and anticipated categories; FN stands for the number of samples where the true category and the forecasted category are both positive; FP stands for the proportion of samples where the anticipated category is positive but the actual category is negative; N=TP+FP+FN+TN is the total number of samples, and TN stands for the number of samples where both the actual and projected categories are negative..

In this paper, 920 images with positive and negative sample labels from the smart car's data set are used to learn and validate lane recognition.According to the confusion matrix above, the accuracy, precision, and recall can be defined. Accuracy is the correct proportion of all predictions and is defined as

            (9)

Precision is correctly predicted as the proportion of Positive which is all Positive and is defined as

                      (10)

Recall is correctly predicted as the proportion of Positive, which is all practically positive, is defined as

       (11)

Another typical metric is the F-measure, which is the weighted average of the precision and recall, defined as

             (12)

The simplified formula is

        (13)

In the formula, when β > 0, it measures the relative importance of recall to precision. When β = 1, it is the standard F1 score, where recall and precision are considered equally important. Furthermore, β > 1 means more emphasis on recall, whereas β < 1 means more emphasis on precision. In our research, our value for β was 1. In our paper,the F1 score combines the results of precision and recall. In this project, the average recall rate of lanes is 87.11%, and the average precision is 89.12% when the calculation of the aforementioned index formulas is also taken into account.

Reviewer 2#:

No further comment. The revision can be accepted now.

Reviewer 3#:

No further comment. The revision can be accepted now.

Thank you very much for your attention and time.Look forward to hearing from you.

Yours sincerely,

Authors
